# A Novel Tissue-Free Method to Estimate Tumor-Derived Cell-Free DNA Quantity Using Tumor Methylation Patterns

**DOI:** 10.3390/cancers16010082

**Published:** 2023-12-23

**Authors:** Collin A. Melton, Peter Freese, Yifan Zhou, Archana Shenoy, Siddhartha Bagaria, Christopher Chang, Chih-Chung Kuo, Eric Scott, Subashini Srinivasan, Gordon Cann, Manami Roychowdhury-Saha, Pei-Yun Chang, Amoolya H. Singh

**Affiliations:** GRAIL, LLC, Menlo Park, CA 94025, USA

**Keywords:** biomarkers, circulating tumor DNA, DNA methylation, computational methods, machine learning algorithms, liquid biopsy, cell-free DNA, tumor fraction, variant allele fraction

## Abstract

**Simple Summary:**

Tumor cells shed pieces of DNA that circulate in biofluids like blood and urine. The amount of circulating tumor DNA is related to the amount of cancer in the body. Knowing the amount of cancer helps doctors predict outcomes and decide treatments for patients. Tumor DNA has methylation patterns that makes it distinct from non-cancer DNA. In this study, we describe a new method called tumor methylated fraction (TMeF) that quantifies the cancer-indicative methylation patterns within circulating tumor DNA from blood samples. This method is non-invasive because it does not need tumor tissue to estimate the amount of cancer in the body. In the future, doctors could use TMeF to supplement current cancer screening methods.

**Abstract:**

Estimating the abundance of cell-free DNA (cfDNA) fragments shed from a tumor (i.e., circulating tumor DNA (ctDNA)) can approximate tumor burden, which has numerous clinical applications. We derived a novel, broadly applicable statistical method to quantify cancer-indicative methylation patterns within cfDNA to estimate ctDNA abundance, even at low levels. Our algorithm identified differentially methylated regions (DMRs) between a reference database of cancer tissue biopsy samples and cfDNA from individuals without cancer. Then, without utilizing matched tissue biopsy, counts of fragments matching the cancer-indicative hyper/hypo-methylated patterns within DMRs were used to determine a tumor methylated fraction (TMeF; a methylation-based quantification of the circulating tumor allele fraction and estimate of ctDNA abundance) for plasma samples. TMeF and small variant allele fraction (SVAF) estimates of the same cancer plasma samples were correlated (Spearman’s correlation coefficient: 0.73), and synthetic dilutions to expected TMeF of 10^−3^ and 10^−4^ had estimated TMeF within two-fold for 95% and 77% of samples, respectively. TMeF increased with cancer stage and tumor size and inversely correlated with survival probability. Therefore, tumor-derived fragments in the cfDNA of patients with cancer can be leveraged to estimate ctDNA abundance without the need for a tumor biopsy, which may provide non-invasive clinical approximations of tumor burden.

## 1. Introduction

Estimates of tumor burden (i.e., the amount of cancer in the body) can guide the clinical management of cancer. Tumor burden is associated with disease outcomes and acts as a prognostic indicator [1,2,3]. As such, it can inform treatment decisions and therapeutic strategies. In addition, response to treatment and disease recurrence can be evaluated by assessing changes in tumor burden over time [4,5]. Yet, measuring tumor burden for broad clinical applications remains a challenge, as current options are mostly limited to radiological imaging. The current standard of care, computed tomography (CT), is relatively accessible but has limitations [6]. For example, detectability is dependent on the size and location of lesions [7], uni- or bi-dimensional measurements of diameter may not correspond to actual tumor volume [5,7,8,9] (with acknowledgment that methods like FDG-PET/CT measure metabolic tumor volume and are being adopted in some clinical practices and trial designs [10,11,12]), the subjective nature of selecting and measuring lesions can lead to variability in tumor measurements [13,14,15], and serial scanning must be balanced with increasing radiation burden [16]. A promising supplemental approach to quantifying tumor burden and capturing tumor growth kinetics is cell-free DNA (cfDNA) analysis.

Biofluids (e.g., blood, urine) from individuals with cancer contain cfDNA derived from both normal cells and cancer cells. Blood levels of cancer-derived cfDNA, termed circulating tumor DNA (ctDNA), have been shown to correlate with tumor volume for multiple cancer types [17,18,19]. Additionally, clinical markers of cancer aggressiveness (e.g., tumor mitotic and metabolic activity, depth of invasion) correlate with ctDNA levels, suggesting that ctDNA analysis captures information on tumor growth, which could better inform risk stratification and prognosis prediction compared to imaging-based assessments of tumor size [20]. Thus, estimating ctDNA abundance in patient blood samples is a promising surrogate measure of tumor burden that can augment imaging. 

A portion of ctDNA fragments contain cancer-associated alleles, which allow ctDNA to be distinguished from normal cfDNA. This fraction of cfDNA in circulation that originates from a tumor and contains cancer-associated alleles is defined as the circulating tumor allele fraction (cTAF) [21]. cTAF is a measurement of ctDNA, and thus approximates overall tumor burden. Higher cTAF, as estimated across a variety of ctDNA detection platforms, has been correlated with worse clinical outcomes [20,21,22,23,24]. 

Current strategies for estimating cTAF largely rely on small variant allele fraction (SVAF) measurements. This approach uses the direct measurement of cfDNA fragments containing small variants, such as somatic single nucleotide variants (SNVs), small insertions, and deletions using digital PCR, quantitative PCR, and error-corrected deep sequencing [25]. To improve sensitivity, this approach is often “tumor-informed,” meaning it requires sequencing of matched tumor samples to identify the small variant alleles subsequently used to detect ctDNA in blood samples [26,27,28]. However, sufficient quantities of tumor tissue are not always available following diagnostic and prognostic testing, and obtaining additional tissue biopsies can be difficult or impossible depending on the tumor location [29,30]. Additionally, these tumor-informed methods are anchored on the primary tumor specimen and can be limited by the clonal diversity and ongoing clonal evolution of the tumor itself [31]. SVAF can be measured with a tissue-free (cfDNA-only; no tumor tissue required) approach, but this is limited by any lack of overlap between tumor-derived variants and the previously defined variants assessed in existing assays. Noise from sources like clonal hematopoiesis of indeterminate potential is also problematic with this approach [32,33], although white blood cell sequencing can be performed as an extra step to remove noise.

DNA methylation patterns provide an alternative, optionally tissue-free means of distinguishing cancer from non-cancer cfDNA to estimate cTAF and approximate tumor burden. Site-specific DNA methylation patterns are characteristic indicators of cellular identity, including neoplastic state [34,35]. The DNA methylation patterns in tumor cells are preserved in ctDNA where they are indicative of cancer presence as well as cancer type [36,37]. Several investigators have evaluated techniques to estimate cTAF from methylation patterns in cfDNA using targeted sequencing, whole-genome bisulfite sequencing (WGBS), and qPCR, with mixed performance [38,39,40,41,42,43,44,45,46]. Most of these methods have not been applied to targeted methylation data, nor have they demonstrated accurate quantification of ctDNA at low concentrations, which is critical for clinical applications (e.g., early cancer detection and minimal residual disease (MRD) monitoring). Here, we describe a novel, sensitive, tissue-free statistical method to quantify cancer-indicative methylation patterns within cfDNA from biofluids to calculate the tumor methylated fraction (TMeF). We robustly demonstrate an order of magnitude improvement in the lower limit of accurate quantification over previously published approaches.

## 2. Materials and Methods

### 2.1. Sample Origin and Prior Processing

Plasma of participants with and without cancer had been previously obtained through the Circulating Cell-Free Genome Atlas (CCGA; NCT02889978) substudy 1 (*n* = 398 non-cancer) [47], CCGA substudy 2 (*n* = 2061 cancer, 1585 non-cancer) [36], and CCGA substudy 3 (*n* = 1434 cancer, 1051 non-cancer) [37]. Inclusion and exclusion criteria for the CCGA substudies have been described previously [36,37]. Briefly, all participants were over 20 years of age and provided written informed consent. Participants with cancer across all clinical stages diagnosed by screening or clinical presentation were enrolled. Participants with current or prior treatment for a diagnosed cancer were excluded. Blood and tissue samples were collected before enrolled participants started definitive therapy. For the substudies, participant samples were selected to meet a pre-specified distribution of cancer types. Formalin-fixed paraffin-embedded biopsy tissues (*n* = 1113) had also been obtained from participants with cancer in the CCGA substudies. Additional tissue and cell type samples had been obtained commercially via Discovery Life Sciences, Huntsville, AL, USA (formerly Conversant Biologics, Inc.). Tissue samples were selected so that each pre-defined cancer label category was represented by at least 15 WGBS tissue samples (note that only the Anus label had less than 15, with 14 sequenced tissue samples; Figure 1b,c).

Tumor sample collection, accessioning, storage, and processing were performed previously as described in the Appendix A provided in Liu et al. 2020 [36]. WGBS of tumor tissue and plasma samples from CCGA substudy 1 was performed previously as described in the Appendix A provided in Liu et al. 2020 [36]. Targeted methylation processing and bisulfite sequencing of plasma samples from CCGA substudies 2 and 3 were performed previously as described in the Appendix A provided in Liu et al. 2020 [36].

### 2.2. Data Processing and Statistical Analysis

We reanalyzed data previously obtained from the CCGA substudies, and R version 4.1.2 (RRID: SCR_001905) was used for all statistical analyses.

### 2.3. Data Processing and Statistical Analysis—Differentially Methylated Region Calling

To distinguish cancer-derived cfDNA from cfDNA shed from non-cancerous cells, we identified a differentially methylated region (DMR) for each cancer-indicative methylation pattern (i.e., a methylation pattern identified from cancer tissue WGBS samples that was hyper- or hypo-methylated relative to non-cancer cfDNA at 5 contiguous CpG sites; Figure 1a). DMRs were identified separately for 1113 CCGA cancer tissue biopsy samples relative to plasma-derived cfDNA WGBS pooled from 398 individuals without cancer from CCGA and defined using data processing and calling thresholds detailed in the Appendix A). Briefly, selected DMRs had cancer-indicative methylation patterns that (1) occurred in less than 0.1% of fragments from the non-cancer plasma WGBS samples (i.e., <0.1% noise) and (2) were present in at least 20% of fragments that contain the contiguous CpG sites of the DMR in at least one cancer sample. 

### 2.4. Data Processing and Statistical Analysis—DMR Clustering

In addition to distinguishing cancer from non-cancer, DMRs can distinguish different cancer types. To identify DMRs for different cancer types, we limited our analysis to regions corresponding to a panel of targeted methylation regions from a previously validated multi-cancer early detection (MCED) test (Galleri^®^, GRAIL, LLC, Menlo Park, CA, USA) for each cancer tissue biopsy sample [36,37]. This MCED test assesses differentially methylated patterns at these targeted regions to provide up to 2 cancer signal origin (CSO) predictions among 20 CSO label options. Similarly, in the present study, the DMRs identified in individual cancer tissue biopsy samples were merged into 20 cancer labels (defined to be consistent with the MCED test CSO labels): Anus, Bladder and Urothelial Tract, Bone and Soft Tissue, Breast, Cervix, Colon and Rectum, Head and Neck, Kidney, Liver and Bile Duct, Lung, Lymphoid Lineage (excluding the Plasma Cell Lineage; this includes both Hodgkin’s and non-Hodgkin’s lymphoma as well as B-cell chronic lymphocytic leukemia and B-cell lymphoblastic leukemia/lymphoma), Melanocytic Lineage (defined here as melanoma of the skin), Myeloid Lineage (includes Acute Myeloid Leukemia and Chronic Myeloid Leukemia), Neuroendocrine Carcinoma (NEC) of Lung or Other Organs, Ovary, Pancreas and Gallbladder, Plasma Cell Lineage, Prostate, Stomach and Esophagus, and Uterus. For each cancer tissue WGBS sample, we calculated the observed frequency of each cancer-indicative methylation pattern occurring within DMRs from all cancer labels. The frequencies within cancer tissue samples of the 50 most prevalent DMRs per cancer label (767 unique DMRs in total) were visualized in a heatmap (Figure 2). DMR prevalence was defined as the expected fraction of cancer tissue samples with a particular cancer label that possesses a particular DMR. Both DMRs and samples within each cancer label were hierarchically clustered using Manhattan distance. Cancer labels were clustered using Spearman’s distance applied to the mean DMR frequency profile across samples of the same cancer label.

### 2.5. Data Processing and Statistical Analysis—DMR Heme Filtering

DMRs for solid cancer labels were additionally filtered to remove any DMRs derived from the hematopoietic lineage by removing overlapping DMRs from custom DMR sets comprising lymphoid lineage, myeloid lineage, and plasma cell lineage. This reduced interference from potentially confounding blood conditions when utilizing DMRs in downstream applications (e.g., cTAF estimation) and filtered DMRs derived from hematopoietic lineage cells resident in tissue biopsy samples. A median of 62% of DMRs were filtered out per solid cancer label (Table 1).

### 2.6. Data Processing and Statistical Analysis—DMR Prevalence Estimation

DMR prevalence was estimated per DMR per cancer label as the median Bayesian posterior estimate of the prevalence given the estimated tumor fractions (TF; i.e., the fraction of cfDNA derived from a tumor) and fragment counts for a set of plasma cfDNA targeted methylation samples labeled with the cancer label of interest (*n* = 2019). Computational details are included in the Appendix A.

### 2.7. Data Processing and Statistical Analysis—Tumor Methylated Fraction Estimation

DMRs were filtered to highly informative non-sex specific regions that had low noise (noise rate below 1/10,000), a strong methylation pattern (i.e., completely methylated or unmethylated), and robust assay performance. The refined DMRs were then used to infer the TF of plasma cfDNA targeted methylation samples from cancer participants by modeling the observed counts of fragments with DMRs as a function of TF (computational details provided in the Appendix A). The TF was converted to an allele fraction, termed TMeF, by multiplying the TF by a scaling factor designed to reflect that a typical cancer-derived small variant allele is heterozygous (Figure 3; Appendix A). Thus, TMeF can serve as an estimate of SVAF. 

### 2.8. Synthetic Dilutions

Pre-treatment plasma cfDNA targeted methylation samples from participants with solid cancers (solid cancer plasma samples) in CCGA substudy 2 [36] and substudy 3 [37] that were held out from TMeF algorithm training and refinement were used for evaluation of TMeF (*n* = 457 cancer and 568 non-cancer). Samples were selected to create a representative subset of cancer types and stages. cfDNA cancer samples with reliably estimated TFs > 0.005 were synthetically mixed with non-cancer cfDNA samples. Each cancer sample was mixed with 3 randomly selected non-cancer samples at the following mixing fractions: r = 3 × 10^−5^, 1 × 10^−4^, 3 × 10^−4^, 1 × 10^−3^, 3 × 10^−3^, 1 × 10^−2^, 3 × 10^−2^, 1 × 10^−1^, 3 × 10^−1^, 1 × 10^0^, forming 3 dilution series per cancer sample with 10 concentrations each. The mixing fractions were post hoc corrected for the difference in coverage between the undiluted cancer and non-cancer samples used in each dilution series using calculations detailed in the Appendix A. Following synthetic dilution, TMeF was estimated for each titrated sample and compared with the theoretical value.

### 2.9. Small Variant Allele Fraction Estimates

Forty-two pre-treatment, solid cancer plasma samples from the CCGA substudy 2 [36] representing 16 different cancer types spanning stages I-IV (16 stage I, 11 stage II, 8 stage III, 7 stage IV samples), an age range of 27 to 85+ (median 63), males (38%) and females (62%), and multiple self-reported ethnicities (2 Asian, Native Hawaiian, or Pacific Islander; 1 Black, non-Hispanic; 2 Hispanic; 2 other/unknown; 35 White, non-Hispanic) were held out from TMeF development and used for SVAF analysis. Samples were selected based on tissue availability and low cancer signal. SVAF estimates were generated using custom targeted panel enrichment and sequencing. Detailed sample processing and data analysis methods are documented in Calef et al. (manuscript in preparation) [48]. Briefly, we used a custom caller designed to identify variants in WGBS data post-bisulfite conversion and whole-genome sequencing of matched cfDNA. Custom targeted panels were then designed for pools of patients with up to 500 small variants selected for each patient. Error-corrected targeted sequencing was performed from cfDNA and read-level data were processed through a custom analysis pipeline to produce counts of fragments with reference and alternate alleles for each set of selected small variants. These counts were used in Bayesian inference to estimate SVAF.

### 2.10. Biophysical Modeling of ctDNA Shedding

Study samples were selected to create a cross-section of solid cancer types and stages and included pre-treatment plasma samples from individuals diagnosed with colorectal, non-small cell lung, breast, prostate, kidney, ovarian, or uterine cancer in CCGA substudy 3 [37] representing clinical stages I-III (*n* = 396; stage IV cancers were excluded to limit the effect of distant metastasis). Tumor size measurements from radiological imaging were extracted from electronic case report forms and summarized as a single maximum tumor size measurement per primary tumor mass. Linear modeling on the log scale was performed to determine scaling of TMeF with tumor size. Robust linear regression was used to confirm the validity of the fit for each model. Details on scaling factor calculations are provided in the Appendix A.

### 2.11. Survival Modeling

Study samples included pre-treatment, solid cancer plasma samples from the CCGA substudy 3 [37] representing clinical stages I-IV (*n* = 1434). The 98th percentile of non-cancer TMeF was used as a cutoff to visually indicate potentially reduced accuracy of TMeF values below this level. One cutoff for each cancer label was empirically determined as the 98th percentile of TMeF values computed on a set of 1051 non-cancer samples. Cutoff values ranged from 6.14 × 10^−5^ to 1.95 × 10^−4^ with a median of 9.80 × 10^−5^. Overall survival was extracted from the study data. Study participants were stratified into 4 groups by their TMeF values (TMeF <10^−4^, 10^−4^–10^−3^, 10^−3^–10^−2^, >10^−2^) and Kaplan–Meier survival curves were generated in R. Cox proportional hazards modeling was used to assess the significance of the association of the TMeF-stratified groups and survival. To provide a reference for survival rates accounting for the heterogeneous mix of sex, ages, clinical stages, and cancer types in the CCGA study, we obtained population-based data of the quarterly overall survival of individuals diagnosed with cancer in 17 regions of the United States from the National Cancer Institute’s Surveillance, Epidemiology, and End Results (SEER) Program and related SEER*Stat program (version 8.4.1). These statistics, which included patients with primary cancer diagnosed between 2006 and 2015 stratified by sex, age (20 years or older to match enrolled CCGA participant ages; 5-year age group), stage at diagnosis (American Joint Committee on Cancer 6th edition stage I, II, III, IV, or unknown), and cancer type (SEER site recode), were adjusted to the CCGA distributions of sex, age, clinical stage, and cancer type. Adjusted SEER data were used to estimate the expected overall survival of the CCGA TMeF-stratified populations, which was compared to the observed overall survival.

## 3. Results

### 3.1. DMRs Are Diverse and Cluster by Cancer Type

By assessing DNA methylation patterns, we identified DMRs to distinguish cancer from non-cancer DNA fragments. As detailed in the Methods, DMRs consist of at least five contiguous CpGs with a single cancer-indicative methylation pattern differentially methylated relative to non-cancer cfDNA (Figure 1a). DMRs were refined to improve signal to noise, which included heme filtering to reduce interference from potentially confounding blood conditions and remove DMRs derived from hematopoietic lineage cells resident in tissue biopsy samples (Table 1). A median of 1911 and 11,683 DMRs were identified per cancer tissue biopsy sample and per cancer label, respectively (Figure 1b). DMRs displayed a wide distribution of prevalence (<1–99%) within each cancer label (Figure 1c). Identified DMRs shared a moderate degree of similarity (<0.6 cosine similarity) between solid cancer labels, with cancer labels of closer biological origins having more overlap in identified DMRs (Appendix A)**.** Visualization of DMR frequencies within cancer tissue biopsy samples via heatmap clustering with the 50 most prevalent DMRs identified per cancer label revealed clusters of both shared and cancer type-specific DMRs (Figure 2). Thus, DMRs may distinguish CSO, consistent with the previously observed high CSO prediction accuracy of the MCED test [37].

### 3.2. TMeF Can Accurately Quantify ctDNA Abundance

One purpose of identifying DMRs is to calculate TMeF from plasma samples without the need for matched tissue sequencing (see Methods). TMeF calculated from plasma cfDNA is a measure of cTAF and overall ctDNA abundance. First, to begin to evaluate the accuracy of TMeF estimations at low ctDNA levels, synthetic dilutions of cancer and non-cancer cfDNA samples were generated at various dilution levels (Figure 4a). At an expected TMeF of 10^−3^, 95% of samples had measured TMeF within 0.5- to 2-fold of the expected TMeF; at an expected TMeF of 10^−4^, it was 77% of samples (Figure 4b,c). TMeF tapers off around 10^−5^; thus, below this level, cTAFs cannot be accurately estimated currently.

Next, we assessed the accuracy of TMeF estimates in 42 pre-treatment, solid cancer plasma samples from the CCGA substudy 2 [36]. Samples represented 16 different cancer types spanning stages I-IV. TMeF estimates were well correlated to SVAF estimates of these samples (Spearman’s correlation of 0.73, *p =* 2.3 × 10^−7^), with 86% of samples (36/42) having a TMeF estimate within 10-fold of the matched SVAF estimate (Figure 5). Of the six samples with a greater than 10-fold discordance in TMeF versus SVAF, four had higher TMeF than SVAF and two had lower TMeF than SVAF. It is important to note that perfect correlation between TMeF and SVAF was not expected due to the limitations of the SVAF measurement. As SVAF is an estimate itself and does not represent the true quantity of ctDNA, it is an imperfect comparator, yet at this time, it is the best method to assess accuracy of the TMeF approach. 

### 3.3. TMeF Is Associated with Clinical Stage and Tumor Size

To probe the relationship between TMeF and tumor burden, TMeF was first computed in a subset of CCGA substudy 3 [37] pre-treatment, solid cancer plasma samples that were held out from algorithm training and refinement. In these samples, TMeF increased with increasing clinical stage both across cancer types (Figure 6a) and within individual cancer types (Appendix A). Additionally, in prostate cancer participant samples stratified by Gleason score, TMeF was higher for participants with scores 4 + 3 and above versus 3 + 4 and below (Appendix A).

TMeF was significantly associated with primary tumor size across a variety of cancer types and subtypes, including colorectal cancer (Appendix A), non-small cell lung cancer (NSCLC) [33] (Appendix A), hormone receptor positive and triple negative breast cancer (TNBC) (Appendix A), and Gleason 4 + 3 and above prostate cancer (Appendix A). Additionally, scaling factors, which represent how tumor shedding relates to tumor size (a scaling factor of 2 or 3 indicates shedding proportional to tumor surface area or volume, respectively), were biologically reasonable (≤3). For kidney cancer (Appendix A), ovarian cancer (Appendix A), and uterine cancer (Appendix A), the scaling factors were small (<1) and not statistically significant, suggesting no association of TMeF and tumor size. This lack of association could indicate that a biological variable other than tumor size is driving the shedding rate in these cancer types. Of note, TNBC had a scaling factor of ~5. This is consistent with the presence of important unmodeled biological variables correlated with tumor size, such as mitotic volume, which was previously found to be a critical factor influencing ctDNA shedding [20]. 

### 3.4. TMeF Is Associated with Overall Survival

Lastly, participants were stratified by TMeF, and those in lower TMeF strata had better overall survival probability compared to participants in higher TMeF strata across cancer types (Figure 6b,c) and within individual cancer types (Appendix A). For all TMeF strata, the observed overall survival of CCGA participants was better than the expected time-dependent overall survival based on SEER populations matched for sex, age, cancer type, and stage. Fold change differences in observed versus expected death rates were greater for lower TMeF strata. In the simple Cox proportional hazard model (Figure 6c), the hazard ratios (HRs) in higher TMeF strata include effects of cancer type and stage due to their correlation with TMeF. Higher TMeF strata are enriched for later-stage cancers, and lower TMeF strata are enriched for earlier-stage cancers (Appendix A), consistent with the observed increase in TMeF with cancer stage (Figure 6a). Nevertheless, lower TMeF was significantly associated with better survival when modeled jointly with clinical stage and cancer type, suggestive of cTAF being prognostic when added to these readily available clinical data (Appendix A).

## 4. Discussion

There is a growing need for estimates of tumor burden to inform the clinical management of cancer. However, the current methods to do so are based on imaging, which is burdensome for patients over time and prone to reader variability [6]. Evidence for ctDNA as a surrogate or supplementary marker of tumor burden is growing [49,50,51,52], but means of easily and effectively measuring ctDNA from biofluid samples are still needed. This paper describes an approach to estimate cTAF using cfDNA methylation patterns and thereby provides an optionally tissue-free means of quantifying tumor burden. Measured TMeF estimates are accurate when compared to the expected TMeF in a synthetic titration series, and TMeF estimates correlate with SVAF estimates of cTAF in the matched plasma samples from patients with cancer. Importantly, TMeF correlates with clinical cancer stage, tumor size, and overall survival, suggesting it may be able to fill the prognostic need for quantifying tumor burden. 

The advantages of using a methylation-based estimate of tumor burden are numerous. First, blood-based methods for cancer detection are less dependent on tumor location and may eventually enable detection before tumors are large enough to be detectable by imaging. Second, methylation patterns can distinguish different cancer types, which could be useful to guide diagnostic imaging when using TMeF in clinical applications of cancer detection and recurrence monitoring. Third, TMeF may capture other clinical markers of cancer aggressiveness and growth, such as tumor mitotic and metabolic activity and depth of invasion, as these markers correlate with ctDNA levels [20], thereby providing better risk stratification and prognosis prediction compared to tumor size alone.

A fourth advantage of TMeF is that ctDNA estimates are expected to be robust across the clonal evolution of tumors among different individuals and within individuals across time because there are many DMRs (typically hundreds to thousands) that are common to each cancer type. In contrast, SNV-based approaches are limited by fewer and patient-specific SNVs [53]. In addition, treatment pressure can contribute to clonal evolution, which can change small variant profiles, making ctDNA less detectable [54]. Due to the broad nature of the signal used to compute TMeF across numerous cancer types, TMeF likely measures ctDNA levels originating from all shedding cancer cells—both from the primary tumor and metastatic sites—regardless of clonal lineage. Lineage transformation is a hallmark of treatment resistance for many cancer types, with cancer cells often transforming to neuroendocrine-like states [55]. We expect methylation patterns of neuroendocrine tumors to emerge, which would be detectable by the TMeF technology; however, additional studies are needed to confirm detectability following lineage transformation. 

Finally, TMeF estimates are reliable in the absence of tumor tissue but retain the flexibility of utilizing matched tissue when tissue samples are available. This is possible because DMRs are both abundant and prevalent within cancer types, making them accurate cancer markers across patient samples without the need for patient-specific, tumor-derived variants to guide detection. Although the tissue-informed approach to SVAF-based cTAF estimates is highly specific, it is limited by tumor tissue quantity, tissue availability, and assay turnaround time. The requirement of obtaining tumor tissue if additional material is needed after diagnostic biopsy places a much greater burden on patients and the healthcare system [56]. Importantly, we confirmed that tissue-free TMeF estimates correlate with tissue-informed SVAF estimates of the same cancer plasma samples. 

We noted a few instances of differences between the SVAF and TMeF metrics, which could be explained by limitations of either the SVAF assay (Calef et al. in preparation [48]) or the methylation assay. Elevated TMeF relative to SVAF could be explained by the presence of tumor DNA shedding with a small variant profile differing from the sampled biopsy, or the existence of multiple health conditions in the participant, such as a second undiagnosed cancer or a pre-malignant heme condition. On the other hand, underestimation of TMeF relative to SVAF could occur if the DNA methylation patterns of the sample of interest differ substantially from what is typically observed in the clinically diagnosed cancer type, or if SVAF estimates are artificially inflated by confounding biological signals (e.g., clonal hematopoiesis of indeterminate potential). For this comparison of TMeF and SVAF estimates, samples were purposefully selected to have a low cancer signal, pushing the limits of TMeF and SVAF assay detection. In samples with higher tumor fractions, the correlation between the two assays would likely improve. 

Other methods to estimate cTAF have been published. For example, copy-number alterations (CNAs) can be used to estimate cTAF without matched tissue genotyping and have been shown to correlate to metastatic prostate cancer prognosis [57]. CNA-based approaches lose accuracy below ~1% tumor fraction, limiting their utility [58], but they can be useful when low limits of detection are not necessary, like evaluating cTAF in advanced disease. Like TMeF, other published methods have focused on methylation patterns in cfDNA to estimate cTAF (Appendix A) [38,39,40,41,42,43,44,45,46]. However, these approaches did not utilize targeted methylation data and have not demonstrated accurate detection at low, clinically meaningful ctDNA levels. These approaches cannot be used with the data in this study without substantial adaptation to account for bias introduced by the targeted pull-down step in our targeted methylation assay. Given the need to generate benchmarking data on an orthogonal platform, SVAF was chosen to assess the accuracy of TMeF estimates given its established low limit of accurate quantification compared to other methylation-based approaches. Although we were not able to perform a head-to-head comparison with existing methylation-based cTAF estimation approaches, from our review of the existing literature, we believe that we are the first to demonstrate a lower limit of accurate tissue-free ctDNA quantification below 0.1% (Appendix A). 

There are limitations to tissue-free approaches, namely the potential for interference from alternative sources of cancer-like molecular signals. For example, clonal hematopoiesis of indeterminate potential can be difficult to distinguish from cancer signals without matched tissue or white blood cell sequencing [32]. Here, we filtered out regions with potential interference from hematopoietic lineages. Another limitation for some tissue-free approaches that do not use methylation biomarkers is that they often sample a limited number of markers, rendering them vulnerable to low ctDNA abundance due to a low fraction of cancer-derived fragments. In contrast, measuring the extent and location of methylation yields many widely spread patterns, allowing accurate quantitation even at low ctDNA abundance. Addressing the challenges of tissue-free estimations of tumor burden facilitates clinical implementation of these methods, which could improve clinical cancer management throughout patient care. 

There were multiple limitations of this study that will be addressed in future work. Here, TMeF linearity upon dilution (to support assessment of TMeF accuracy at low ctDNA levels) was determined by in silico analysis. Although TMeF linearity was consistent with previously reported results from in vitro dilution assays that used TMeF to assess the limit of detection of a post-diagnosis cancer detection test [59], future validation of TMeF with in vitro cfDNA dilutions will be performed to more fully assess its linearity. It should also be noted that there are epigenetic differences within the cancer label groupings used in this study. An important challenge in the presented work was the balance between choosing cancer label definitions that capture cancer types with shared methylation patterns while also having sufficient sample numbers to capture both the diversity and prevalence of these methylation patterns.

Additionally, TMeF will be further validated in diverse populations of clinical subjects including different ages, sexes, races/ethnicities, cancer types, and cancer stages. TMeF relies on DMR prevalence estimates defined for specific cancer labels and is expected to be most accurate in subjects with cancer methylation patterns most closely aligned with those of the corresponding training set for the given cancer label. Future work will be needed to measure the effect of methylation pattern heterogeneity from genetic ancestry, cancer subtypes, and cancer clonal evolution on TMeF accuracy. Preliminary analysis of methylation patterns across different self-reported ethnicities suggests that methylation pattern differences by ancestry would not have a significant effect on assay performance [60]. 

Lastly, the analysis of TMeF prognostic power within stage and cancer type is limited here by the small sample sizes for specific stage and cancer type groups. We noted that in cases where TMeF had a weaker correlation to tumor size, it was for cancer types such as kidney, ovarian, and uterine, which are known to have low rates of shedding into the bloodstream [61,62] and may not be detected given the current lower limit of TMeF at approximately 10^−5^. We also noted that CCGA participants had better survival relative to SEER-expected survival in each TMeF stratum, which could be explained by a healthy volunteer effect and this population’s access to medical care [24].

TMeF has many possible clinical applications. The TMeF technology described here complements the methylation-based machine learning classifier used by the GRAIL MCED test. The MCED test detects a shared cancer signal within cfDNA methylation patterns and provides a binary cancer signal ”detected” or ”not detected” output along with a CSO prediction [36,37]. TMeF, an independent algorithm, provides a means of quantifying the cancer signal, allowing estimations of tumor burden. The association of TMeF and survival probability supports the use of TMeF to stratify clinical trial participants by identifying those at high risk of recurrence [33]. Recent studies to develop a prognostic test for early-stage lung adenocarcinoma (LUAD) have applied a methylation-based machine learning classifier, for which ctDNA detection is well correlated with TMeF, to assess ctDNA status from blood samples [63,64]. Pre-surgical ctDNA detection in stage I LUAD was associated with worse recurrence-free survival and overall survival [64]. With further development, this prognostic test may be useful to guide clinical decision making and identify high-risk participants for clinical trials. 

At the patient level, TMeF could be used as a complement to imaging to provide a baseline evaluation of tumor burden at the start of cancer treatment. Tumor burden at baseline can act as a prognostic indicator, with high tumor burden associated with worse outcomes. For example, Chabon et al. showed that low cTAF levels in pre-treatment, early-stage NSCLC were significantly associated with decreased risk of recurrence [22]. Furthermore, tumor burden at baseline may act as a predictive marker for certain therapies. For example, high tumor burden may indicate immune checkpoint inhibitors are less likely to be effective [3]. Additionally, TMeF estimates of tumor burden could be useful during treatment to enable tracking of tumor kinetics in response to therapy and potentially act as a predictive biomarker. Nabet et al. demonstrated that a decreasing ctDNA trajectory after a single cycle of immune checkpoint inhibition treatment in NSCLC was significantly associated with improved progression-free survival [23]. Finally, TMeF could be used in MRD applications after treatment completion. Multiple studies have demonstrated the association of post-treatment ctDNA detection with worse prognosis [65,66]. As a sensitive tissue-free measure of ctDNA abundance, TMeF could act as a prognostic marker in the MRD setting. In addition to clinical applications, TMeF is a potential metric for analytical validation studies, due to its ease of implementation and accuracy, specifically to assess the abundance of an analyte in sensitivity (limit of detection) and specificity studies [59]. Studies investigating the use of TMeF in these clinical applications are underway.

## 5. Conclusions

In summary, TMeF is a broadly useful metric applicable to the study of ctDNA abundance in biofluids. In this work, we demonstrated that TMeF, derived from plasma samples of patients with cancer, can be used as an estimate of cTAF without need for matched tumor samples. Further studies are needed to validate TMeF as a clinical measure of cTAF and tumor burden. Beyond reflecting tumor volume, TMeF may capture additional tumor biology (mitotic and metabolic activity, depth of invasion) that could be used to predict tumor aggressiveness [20]. In the future, TMeF estimates of ctDNA can be applied to other biofluids such as urine, and it may be used for a variety of clinical applications, such as early cancer detection, prognosis predictions, MRD, and recurrence monitoring.

## Figures and Tables

**Figure 1 cancers-16-00082-f001:**
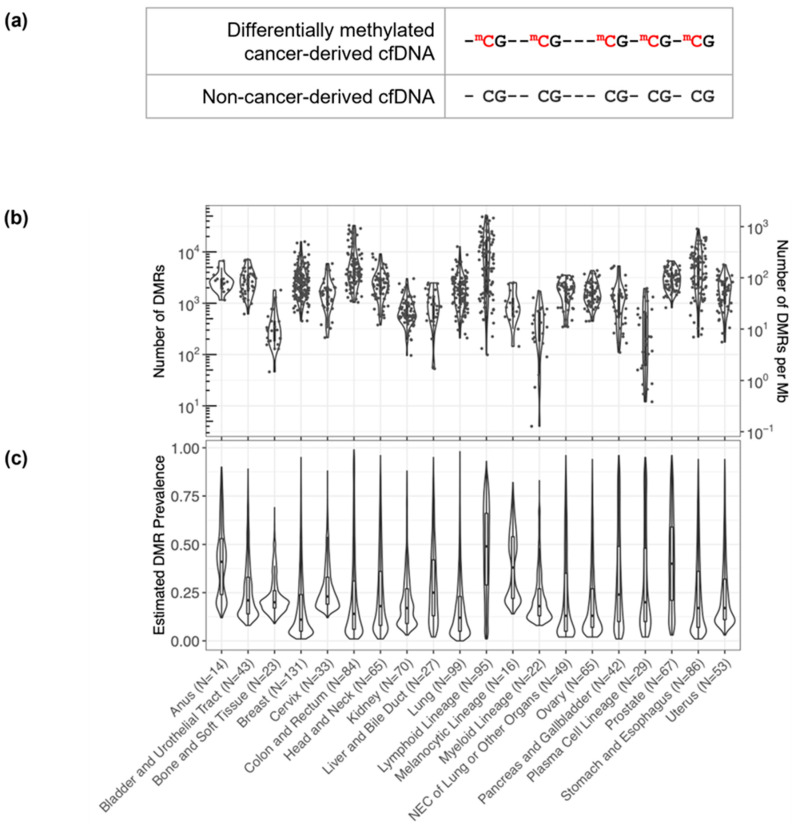
Discovery of short DMRs in cancer tissue samples. (**a**) A pictorial representation of a short 5 CpG DMR highlighting the differential methylation pattern in the cancer-derived pre-treatment cfDNA (red) relative to non-cancer-derived cfDNA. (**b**) Within each cancer sample, hundreds to thousands of DMRs were identified. The number of DMRs identified per sample per cancer label is plotted and overlaid as a violin plot summarizing the distribution across the samples. (**c**) For each DMR identified within each cancer label, the prevalence of the DMR (i.e., the fraction of cancer tissue samples in which the DMR occurs) per cancer label was estimated. The distributions of the DMR prevalence estimates per cancer label are each displayed as a violin plot overlaid with a box plot. N values correspond to the number of cancer tissue samples per cancer label.

**Figure 2 cancers-16-00082-f002:**
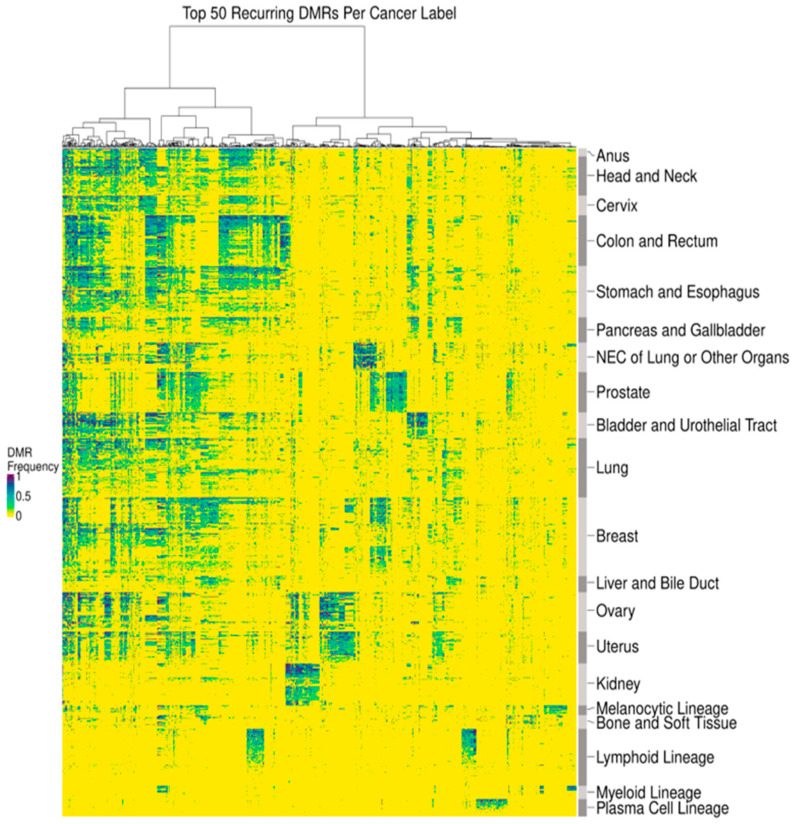
DMRs delineated cancer type-associated methylation patterns. A heatmap depicting the observed DMR frequency of the 50 most prevalent DMRs per cancer label (x-axis) across tissue samples (y-axis). Samples within each cancer label were clustered using Manhattan distance, and cancer labels were clustered using Spearman’s distance applied to a per cancer label average. DMRs were clustered by Manhattan distance.

**Figure 3 cancers-16-00082-f003:**
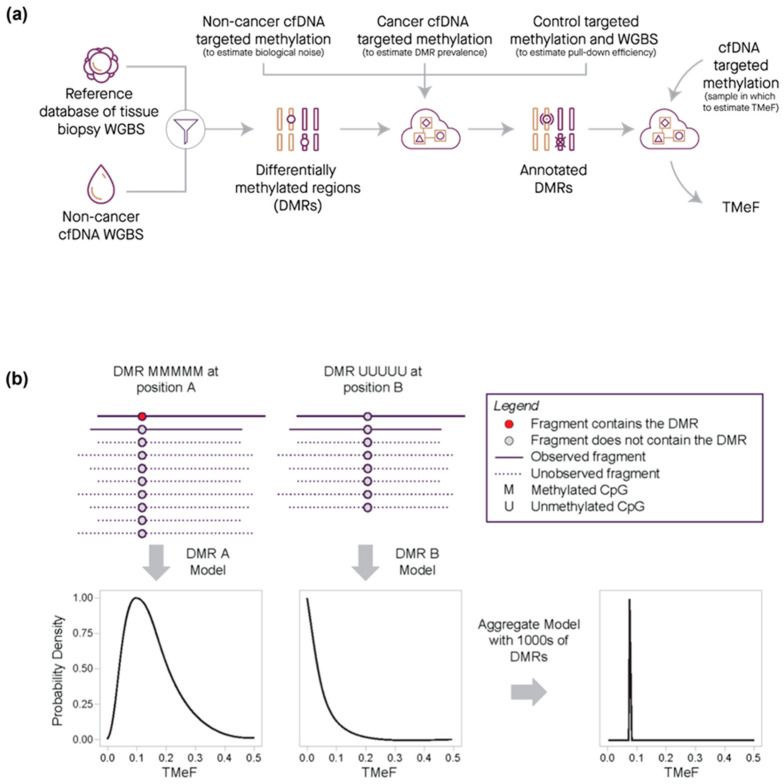
Quantification of TMeF by comparing DMR cancer-indicative methylation patterns. (**a**) Schematic of the information flow for generating a TMeF estimate. First, DMRs were identified that differentiate tissue biopsy methylation WGBS for a particular cancer type from non-cancer cfDNA WGBS. Next, DMRs were annotated with information derived from non-cancer cfDNA targeted methylation (an estimate of biological noise), cancer cfDNA targeted methylation (an estimate of DMR prevalence), and both control sample targeted methylation and WGBS (an estimate of pull-down efficiency). Finally, counts of fragments with annotated DMRs in a targeted methylation cfDNA sample were used to estimate TMeF. (**b**) Schematic illustrating the TMeF computation. Sequenced fragments are shown as solid lines with DMRs (red dots) and without DMRs (gray dots) at 2 genomic sites (position A and position B). Unobserved fragments both with and without DMRs are depicted by dashed lines. Site specific counts of fragments with DMRs are consistent with a range of possible TMeF levels with each level conveying a specific likelihood. Aggregation of likelihoods across loci results in a sample-level TMeF estimate.

**Figure 4 cancers-16-00082-f004:**
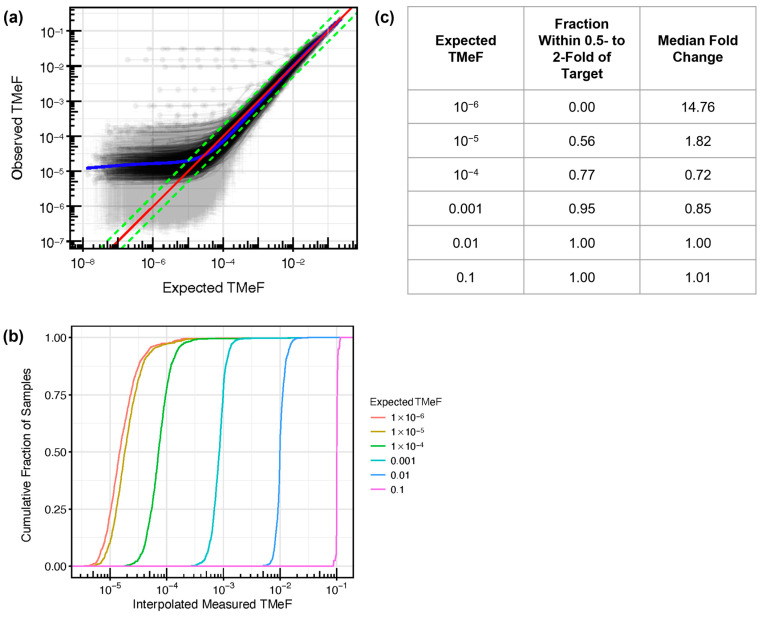
Synthetic dilution analysis assessed TMeF linearity. (**a**) Synthetic dilutions were generated by mixing each of 457 pre-treatment, solid cancer cfDNA samples from CCGA substudy 3 into a paired randomly matched non-cancer cfDNA background sample. Dilutions were generated in triplicate across a series of dilution levels, and the measured TMeF was plotted against the expected TMeF. The red line indicates y = x (i.e., expected TMeF = observed TMeF). The green lines represent y = 0.5x and y = 2x (in log space this results in a difference in intercept) as a visual reference for how many curves are within 0.5- to 2-fold of the target. The blue line shows the best fit as determined using a general additive model with the restricted maximum likelihood method. A small number of outlier series can be seen with high observed TMeF across all dilution levels. This is due to the high level of background signal in the specific matched non-cancer samples used in each of these cases. (**b**) For vertical slices in (**a**) at fixed expected TMeF values, the cumulative fraction of observed TMeF was interpolated and plotted. (**c**) For each cumulative distribution in (**b**) at a fixed expected TMeF value, the fraction of measured TMeF within 0.5- to 2-fold of the expected TMeF and the median fold-change deviation from the expected TMeF were calculated. The expected TMeF values include 10^−6^ to demonstrate the limited TMeF accuracy at this low ctDNA level.

**Figure 5 cancers-16-00082-f005:**
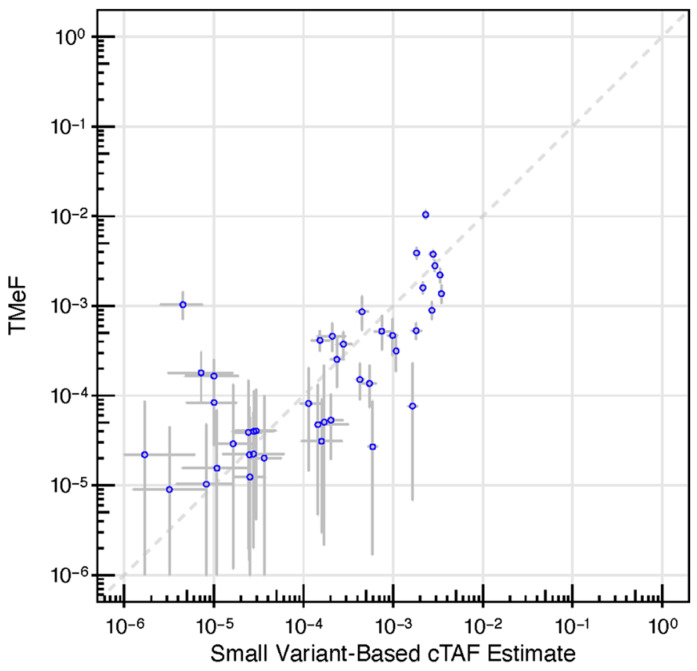
DMRs enabled allele fraction estimation. A scatter plot depicting TMeF (y-axis) vs. patient-specific panel small variant estimates (x-axis) in pre-treatment plasma samples from CCGA substudy 2 participants with solid cancers. TMeF and SVAF estimates correlated with a Spearman’s correlation of 0.73, *p =* 2.3 × 10^−7^. Points indicate posterior median. Error bars represent the 95% credible interval defined by the 2.5 and 97.5 percentiles of the posterior allele fraction distribution.

**Figure 6 cancers-16-00082-f006:**
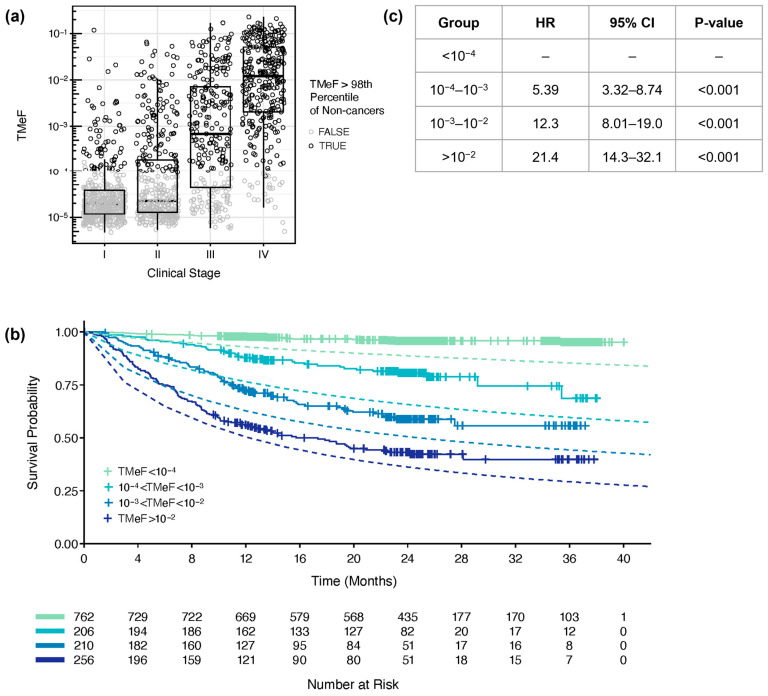
TMeF correlated with clinical stage and survival. (**a**) TMeF for 1434 pre-treatment plasma samples from CCGA substudy 3 participants with solid cancers is plotted against clinical stage. Points are colored gray if the sample’s TMeF was lower than the 98th percentile of TMeFs computed on a set of 1051 non-cancer samples to indicate that these TMeF values were less accurate. TMeF and stage correlated with a Spearman’s correlation of 0.65, *p* = 1.2 × 10^−173^. (**b**) In total, 1434 solid cancer participant plasma samples were stratified by their TMeF, and Kaplan–Meier plots of overall survival were generated for each stratified set of participants. Dashed lines depict the expected time-dependent overall survival based on SEER populations matched for sex, age, cancer type, and stage for each TMeF stratum. (**c**) The Cox proportional hazards model HRs and *p*-values were calculated for TMeF-stratified participant groups. Caveats of the model and associated HRs are described in the Results.

**Table 1 cancers-16-00082-t001:** Number of DMRs within each cancer label. DMRs identified in solid cancer labels were filtered to remove DMRs identified in the plasma cell lineage, lymphoid lineage, or myeloid lineage (i.e., “heme filtering”). The number of DMRs before and after filtering as well as the fraction removed are depicted.

Number of DMRs in Each Cancer Label			
Cancer Label	Unfiltered DMRs	Heme-Filtered DMRs	Fraction Removed
Anus	27,804	10,176	0.63
Bladder and Urothelial Tract	39,587	16,278	0.59
Bone and Soft Tissue	7389	3875	0.48
Breast	76,882	32,916	0.57
Cervix	30,740	10,397	0.66
Colon and Rectum	111,576	47,363	0.58
Head and Neck	51,231	21,965	0.57
Kidney	15,173	7542	0.5
Liver and Bile Duct	21,186	7856	0.63
Lung	66,613	28,583	0.57
Lymphoid Lineage	80,078	80,078	0
Melanocytic Lineage	11,559	5111	0.56
Myeloid Lineage	4876	4876	0
NEC of Lung or Other Organs	28,289	13,716	0.52
Ovary	30,070	14,800	0.51
Pancreas and Gallbladder	33,631	10,262	0.69
Plasma Cell Lineage	3558	3558	0
Prostate	25,525	11,696	0.54
Stomach and Esophagus	99,424	41,219	0.59
Uterus	30,335	11,671	0.62

## Data Availability

All relevant data are within the paper and the GitHub public repository. Summary data tables and code required to generate all figures and tables are available at https://github.com/grailbio-publications/tmef2023 (to be made public upon publication).

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
