# Peer review of "A Novel Tissue-Free Method to Estimate Tumor-Derived Cell-Free DNA Quantity Using Tumor Methylation Patterns"

_cancers, 2023, doi:10.3390/cancers16010082_

Round 1

Reviewer 1 Report

Comments and Suggestions for Authors

Dear Editor,

I have reviewed the paper by Melton C.A., et al., titled: A Novel Tissue-Free Method to Estimate Tumor-Derived Cell-2 Free DNA Abundance Using Tumor Methylation Patterns. The study is based on a statistical method to quantify cancer methylation patterns within cfDNA to estimate ctDNA abundance. The study identified differentially methylated regions between a reference cancer tissue biopsy samples and cfDNA from individuals without cancer. Additionally,  differentially methylated regions were used to determine a tumor methylated fraction to quantify circulating tumor allele fraction and estimate of ctDNA abundance. In general, the study could be interesting for the journal, however, it is necessary to make modifications before its possible publication. Below I detail my comments:

Title:

The study is based on the quantification of cfDNA, so the word quantity would be more descriptive instead of abundance.

Introduction:

The introduction is extremely long, an important space is lost developing known explanations about the value of cfDNA. This part should be reduced and contextualized. Additionally, calls to supplementary material should be eliminated.

Material and Methods:

The inclusion of different types of cancer, as well as different tumor stages, makes it difficult to understand how the samples were chosen. It is necessary to explain the inclusion and exclusion criteria used in the study. One of the tumors with the greatest amount of genomic alterations is melanoma. What does the term melanocytic lineage refer to? And likewise, what do the terms myeloid and lymphoid lineages refer to?. 

Within the groups indicated there are important genomic and therefore also epigenetic differences.

Discussion:

The discussion should be focus on the comparison with previous studies. It is necessary to highlight the limitations of the study in a paragraph. It is not necessary to call figures or images, since they should have been presented in the results.

Minor changes:

-       Since the study focuses on cfDNA, biofluids should be changed to cfDNA.

-       Line 46: detectable [7]. point should be eliminated.

Comments on the Quality of English Language

English is fine.

Author Response

Reviewer 1

  • The study is based on the quantification of cfDNA, so the word quantity would be more descriptive instead of abundance.

We agree with the reviewer and have updated the title accordingly (“A Novel Tissue-Free Method to Estimate Tumor-Derived Cell-Free DNA Quantity Using Tumor Methylation Patterns”).

  • The introduction is extremely long, an important space is lost developing known explanations about the value of cfDNA. This part should be reduced and contextualized. Additionally, calls to supplementary material should be eliminated.

We appreciate the reviewer’s note about the length and focus of the Introduction. We have substantially revised the Introduction to focus on the most relevant background information and have reduced details on the known value of cfDNA.

As suggested by the reviewer, we also removed the callout to Figure S1 in the Introduction. This figure is now Figure S13 and is referenced in the Discussion (line 467 and 477).

  • The inclusion of different types of cancer, as well as different tumor stages, makes it difficult to understand how the samples were chosen. It is necessary to explain the inclusion and exclusion criteria used in the study. One of the tumors with the greatest amount of genomic alterations is melanoma. What does the term melanocytic lineage refer to? And likewise, what do the terms myeloid and lymphoid lineages refer to? Within the groups indicated there are important genomic and therefore also epigenetic differences.

We thank the reviewer for calling out a lack of clarity regarding sample selection in the Methods section. We have added details on the inclusion and exclusion criteria for the CCGA study in the “Sample origin and prior processing” subsection (lines 104-116). Additionally, we clarified the rationale for sample selection used for synthetic dilutions (lines 231-232), SVAF analysis (lines 248), and shedding and survival modeling (lines 259 and 270) within the respective subsections.

In response to the reviewer’s suggestion, we have also clarified the definitions of melanocytic, myeloid, and lymphoid lineages in this study within the “DMR clustering” subsection (lines 161-165). We agree with the reviewer’s comment that there are epigenetic differences within our cancer label groupings. An important challenge in the presented work was the balance between choosing cancer label definitions that capture cancer types with shared methylation patterns while also having large enough sample numbers to capture both the diversity and prevalence of these methylation patterns. We have added additional detail in the Discussion to highlight this limitation (lines 496-500).

  • The discussion should be focus on the comparison with previous studies. It is necessary to highlight the limitations of the study in a paragraph. It is not necessary to call figures or images, since they should have been presented in the results.

We agree with the reviewer regarding the focus and content of the Discussion. We have eliminated references to figures discussed in the Results and have removed text that was redundant to the Results. We also reorganized some of the Discussion paragraphs for clarity. As suggested by the reviewer, we now have a full paragraph that discusses this work in comparison to other methods of cTAF estimation (starting at line 461). Furthermore, Figure S13 (originally Figure S1) visually summarizes the comparison of TMeF to other published methylation-based ctDNA assays and is referenced in the Discussion (line 467 and 477). We also added additional limitations of the study, which are now described in 3 paragraphs (starting at line 490).

  • Since the study focuses on cfDNA, biofluids should be changed to cfDNA.

We thank the reviewer for this suggestion to replace “biofluids” with “cfDNA.” In the context of this manuscript, biofluid is not used interchangeably with cfDNA and is only used to clarify the source of the cfDNA used for TMeF analysis. We feel this is an important point to maintain as biofluid-derived cfDNA is fundamental to the TMeF method. Being tissue-free, TMeF is trained with data from the biofluid of interest (blood in this study, but additional studies are investigating urine-derived cfDNA [Stewart T, Shenoy A, Stuart SM, et al. Urine cell-free DNA improves detection of bladder cancer compared to blood-based screening. Presented at the 2023 ASCO Genitourinary Cancers Symposium; February 16 - 18, 2023; San Francisco, CA. https://grail.com/wp-content/uploads/2023/02/Stewart_ASCO-GU-2023_Urine-cfDNA-Bladder-Cancer-Detection_Poster_FINAL.pdf]).

  1. Line 46: detectable [7]. point should be eliminated.

We appreciate the reviewer’s notice of this error. This sentence was deleted with the reviewer’s suggested revision of the Introduction.

Reviewer 2 Report

Comments and Suggestions for Authors

This paper describes a novel tissue-free method for estimating tumor-derived cell-free DNA (cfDNA) abundance using tumor methylation patterns. The researchers developed a statistical algorithm that quantifies cancer-indicative methylation patterns within cfDNA extracted from biofluids, allowing for the estimation of circulating tumor DNA (ctDNA) abundance without needing a tumor biopsy.

This paper provides adequate results to support their hypothesis and conclusion; the paper is suitable to be published. Here are some suggestions and questions for the authors:

  1. In Figure 2, consider moving the legend to the left of the chart.
  2. How is the method applicable to a wider demographic distribution? Are there variations in methylation patterns across different populations that might affect the accuracy of the algorithm? How well does the method perform across diverse demographic groups, including different ages, sexes, ethnicities, and stages of disease?

Author Response

Reviewer 2

  • In Figure 2, consider moving the legend to the left of the chart.

We agree with the reviewer and have moved the Figure 2 key to the left-side of the heatmap (line 176).

  • How is the method applicable to a wider demographic distribution? Are there variations in methylation patterns across different populations that might affect the accuracy of the algorithm? How well does the method perform across diverse demographic groups, including different ages, sexes, ethnicities, and stages of disease?

We greatly appreciate these important questions raised by the reviewer. For clarity, we added demographic information for the study samples used for SVAF comparison analysis to the SVAF subsection of the Methods (lines 244-246). These samples spanned 16 cancer types, stages I-IV (16 stage I, 11 stage II, 8 stage III, 7 stage IV samples), an age range of 27 to 85+ (median 63), males (38%) and females (62%), and multiple self-reported ethnicities (2 Asian, Native Hawaiian, or Pacific Islander; 1 Black, non-Hispanic; 2 Hispanic; 2 other/unknown; 35 White, non-Hispanic). This suggests that the TMeF method performs well across demographic groups.

The reviewer points out an important limitation of our study; the TMeF method will have to be validated in diverse populations that include different ages, sexes, races/ethnicities, cancer types, and cancer stages. We have added this to the study limitations paragraph of the Discussion (lines 501-510). We do believe this method will be applicable to a wider demographic distribution, as has been demonstrated for GRAIL classifiers. Studies of classifiers built on the same targeted methylation assay utilizing samples from individuals with different races and ethnicities have shown that genetic ancestry does not significantly affect performance (Venn O, Bredno J, Thornton A, et al. Robustness of a targeted methylation-based multi-cancer early detection (MCED) test to population differences in self-reported ethnicity. Presented at the 2023 AACR Conference on The Science of Cancer Health Disparities in Racial/Ethnic Minorities and the Medically Underserved; September 29-October 2, 2023; Orlando, FL. https://grail.com/wp-content/uploads/2023/09/Venn_AACR-SCHD-2023_TM-by-RE_Poster_FINAL.pdf). We also added this point and citation to the Discussion (lines 508-510).

Reviewer 3 Report

Comments and Suggestions for Authors

The manuscript focuses on the identification of a novel methylation based assay for the evaluation of tumor derived nucleic acids in solid tumor patients. Accordingly, the manuscript is timely relevant and well structured and may be accepted for the publication on this journal after few minor comments

- In the methodological section, please, could the authors show the cut off for TMeF high and low adopted in the text? Could it was empirically accepted or experimentally acquired?

- In the discussion section, please, could the authors design a diagnostic alghoritm where this approach may be available in clinical practice?

Comments on the Quality of English Language

Minor english editing

Author Response

Reviewer 3

  • In the methodological section, please, could the authors show the cut off for TMeF high and low adopted in the text? Could it was empirically accepted or experimentally acquired?

We appreciate the reviewer’s suggestion to improve clarity of TMeF cutoffs used in the study within the Methods section. Accordingly, we have added detail on the empirically determined 98th percentile non-cancer TMeF cutoff used to visually indicate potentially lower accuracy of TMeF values below this level (lines 271-275). Additionally, we added the TMeF values of the stratified groups used in the survival analysis (lines 276-277).

  • In the discussion section, please, could the authors design a diagnostic alghoritm where this approach may be available in clinical practice?

We thank the reviewer for this suggestion and agree that it is important to discuss the potential future clinical applications of the TMeF method. As such, we have added an example of a prognostic test for lung adenocarcinoma that is currently being developed (lines 526-532). This test uses a methylation-based machine learning classifier to determine ctDNA status, which is associated with worse outcomes and could, in the future, be applied to clinical practice to inform decision making or stratify participants in clinical trials. TMeF (referred to as methyl ctDNA level in this study) is shown to be highly correlated with both classifier status and clinical outcomes (Hong TH, Hwang S, Abbosh C, et al. Tumor-naïve pre-surgical ctDNA detection is prognostic in stage I lung adenocarcinoma, associating with PD-L1 positivity and high-grade histological subtype. Presented at the 2023 North American Conference on Lung Cancer; December 1-3, 2023; Chicago, IL. [poster PDF provided as non-published material for reviewers]). Ongoing and future studies are investigating other clinical applications of TMeF like an MRD test. We currently discuss these future clinical applications of TMeF in the Discussion beginning at line 519.